# Hessian Eigenspectra of More Realistic Nonlinear Models

**Zhenyu Liao**[*†]
School of Electronic Information and Communications
Huazhong University of Science and Technology, China
`zhenyu_liao@hust.edu.cn`

**Michael W. Mahoney**
ICSI and Department of Statistics
University of California, Berkeley, USA
`mmahoney@stat.berkeley.edu`

## Abstract

Given an optimization problem, the Hessian matrix and its eigenspectrum can be used in many ways, ranging from designing more efficient second-order algorithms to performing model analysis and regression diagnostics. When nonlinear models and non-convex problems are considered, strong simplifying assumptions are often made to make Hessian spectral analysis more tractable. This leads to the question of how relevant the conclusions of such analyses are for realistic nonlinear models. In this paper, we exploit tools from random matrix theory to make a *precise* characterization of the Hessian eigenspectra for a broad family of nonlinear models that extends the classical generalized linear models, without relying on strong simplifying assumptions used previously. We show that, depending on the data properties, the nonlinear response model, and the loss function, the Hessian can have *qualitatively* different spectral behaviors: of bounded or unbounded support, with single- or multi-bulk, and with isolated eigenvalues on the left- or right-hand side of the main eigenvalue bulk. By focusing on such a simple but nontrivial model, our analysis takes a step forward to unveil the theoretical origin of many visually striking features observed in more realistic machine learning models.

## 1   Introduction

The Hessian is ubiquitous in applied mathematics, statistics, and machine learning (ML). Given a (loss) function $L(\mathbf{w})$ of some parameters $\mathbf{w} \in \mathbb{R}^p$, the Hessian $\mathbf{H}(\mathbf{w}) \in \mathbb{R}^{p \times p}$ is defined as the second derivative of the loss function with respect to the model parameter, i.e., $\mathbf{H}(\mathbf{w}) = \partial L(\mathbf{w})/(\partial \mathbf{w} \partial \mathbf{w}^\mathsf{T})$. When a ML model is being trained, it is common to parameterize that model by $\mathbf{w}$, and then train that model by minimizing some (smooth) loss function $L(\mathbf{w})$, with the associated Hessian $\mathbf{H}(\mathbf{w})$, e.g., by backpropagating the error to improve $\mathbf{w}$ [26]. Alternatively, once a ML model is trained, the Hessian (and the related Fisher information matrix [66, 68]) can be examined to identify outliers, perform diagnostics, and/or engage in other sorts of model validation [31, 79, 62].

For convex problems, the Hessian $\mathbf{H}(\mathbf{w})$ provides detailed information on how to adjust the gradient to achieve improved convergence, e.g., in Newton-like methods. For non-convex problems, the properties of the local loss "landscape" around a given point $\mathbf{w}$ in the parameter space is of central

---

[*]Work done at ICSI and Department of Statistics, University of California, Berkeley, USA.

[†]We refer the readers to an extended version of this article [43] for detailed proofs and more discussions.

35th Conference on Neural Information Processing Systems (NeurIPS 2021).

significance [18, 36, 13, 39, 77, 78, 79]. In this case, most obviously, the signs of the smallest and largest Hessian eigenvalue can be used to test whether a given $\mathbf{w}$ is a local maximum, local minimum, or a saddle point. More subtly, the Hessian eigenvalue distribution characterizes the local curvature of the loss function and provides direct access to, for instance, the fraction of negative Hessian eigenvalues that determines the number of (local) descent directions, a quantity that is directly connected to the rates of convergence of various optimization algorithms [33].

For theoretical analysis of neural network (NN) models, Hessian eigenspectra are often assumed to follow well-known random matrix distributions such the Marčenko–Pastur law [45] or the Wigner's semicircle law [70]. This enables one to use Random Matrix Theory (RMT), but it involves (for NNs, at least) making relatively strong simplifying assumptions (e.g., the Hessian can be decomposed as the sum of the two freely independent matrices, the residual error, data feature, and weights are all composed of i.i.d. zero mean normal random variables) [57, 58, 15]. A somewhat more realistic setup involves using a so-called *spiked model* (or a spiked covariance model) [3, 5, 41]. In this case, the matrix follows a *signal-plus-noise* model and consists of *full rank* random noise matrix and *low rank* statistical information structure.[3] The "signal" eigenvalues are generally larger than the noisy "bulk" eigenvalues; and the maximum eigenvalues, when isolated from the bulk, are referred to as the "spikes." A substantial theory-practice gap exists, however. In both toy examples [27] and state-of-the-art NN models [77, 78, 79, 80, 64, 20], the strong simplifying assumptions are far from satisfactory. (A similar theory-practice gap has been observed for other NN matrices to which RMT has been applied, perhaps most notably weight matrices [47, 48, 46].) A more precise understanding of the Hessian eigenspectra (and its dependence on the input data structure, the underlying response model and model parameters, as well as the loss function) for more realistic models is needed.

## 1.1 Our approach

In this article, we address these issues, in a setting that is simple enough to be analytically tractable but complex enough to shed light on realistic large-scale models. We consider a family of generalized generalized linear models (G-GLMs) that extends the popular generalized linear model (GLM) [19, 31]; and we show that, even for such simple models, the key simplifying assumptions used in previous theoretical analyses of Hessian can be very inexact. In particular, apart from a few special cases (including linear least squares and logistic regression with homogeneous features), most Hessians of G-GLMs are *not* close to the Marčenko–Pastur and/or the semicircle law. Instead, the corresponding Hessian depends on the input feature structure, the underlying response model, and the loss function, in a more involved fashion that can be precisely characterized by the proposed analysis.

The G-GLM describes a generalized linear relation between the input feature $\mathbf{x}_i \in \mathbb{R}^p$ and the corresponding response $y_i$, in the sense that there exists some parameters $\mathbf{w}_* \in \mathbb{R}^p$ such that for given $\mathbf{w}_*^\mathsf{T}\mathbf{x}_i$, the response $y_i$ is independently drawn from

$$y_i \sim f(y \mid \mathbf{w}_*^\mathsf{T}\mathbf{x}_i) \tag{1}$$

for some conditional density function $f(\cdot \mid \cdot)$. This extends the classical GLM such as

$$\text{logistic model: } \mathbb{P}(y = 1 \mid \mathbf{w}_*^\mathsf{T}\mathbf{x}) = (1 + e^{-\mathbf{w}_*^\mathsf{T}\mathbf{x}})^{-1}, \quad y \in \{-1, 1\}, \tag{2}$$

and covers a large family of models in applications in statistics and ML. Other examples include: (i) the (noisy) nonlinear factor model where $y \sim \mathcal{N}(g(\mathbf{w}_*^\mathsf{T}\mathbf{x}), \sigma^2)$ for some nonlinear $g : \mathbb{R} \to \mathbb{R}$ and $\sigma > 0$ [14]; (ii) the (noiseless) phase retrieval model with $y = (\mathbf{w}_*^\mathsf{T}\mathbf{x})^2$, in which case one wishes to reconstruct $\mathbf{w}_*$ from its (squared) magnitude measurements [21]; and (iii) the single-layer NN model $y = \sigma(\mathbf{w}_*^\mathsf{T}\mathbf{x})$ for some nonlinear activation function $\sigma(t)$ such as the tanh-sigmoid $\sigma(t) = \tanh(t)$.

For a given training set $\{(\mathbf{x}_i, y_i)\}_{i=1}^n$ of size $n$, the standard approach to obtain/recover the parameter $\mathbf{w}_* \in \mathbb{R}^p$ of a G-GLM is to solve the following optimization problem

$$\min_{\mathbf{w}} L(\mathbf{w}) = \min_{\mathbf{w}} \frac{1}{n}\sum_{i=1}^n \ell(y_i, \mathbf{w}^\mathsf{T}\mathbf{x}_i), \tag{3}$$

for some loss function $\ell(y, h) : \mathbb{R} \times \mathbb{R} \to \mathbb{R}$, e.g., the negative log-likelihood of the observation model within the maximum likelihood estimation framework [31] such as the logistic loss $\ell(y, h) =$

---

[3]Since the *same* informative pattern is repeated in each row or column of the matrix.

$\ln(1 + e^{-yh})$ in the case of logistic model (2). In many applications, however, the optimization problem in (3) may *not* be convex, for example to achieve superior robustness and/or accuracy [49, 71, 10], and can be NP-hard in general (the noiseless phase retrieval model $y = (\mathbf{w}_*^\mathsf{T}\mathbf{x})^2$ with the square loss $\ell(y, h) = (y - h^2)^2$ as an example [9]). As we shall see, in such non-convex G-GLMs, the dominant Hessian eigenvector can be shown, in some cases, to positively correlate with the sought-for parameter $\mathbf{w}_*$ and therefore be used as the initialization of gradient descent methods [9, 37, 34]. This particularly motivates our study of the possible isolated Hessian eigenvalue-eigenvector pairs.

## 1.2 Our main contributions

The main contribution of this work is the *exact* characterization of Hessian eigenspectra for the family of G-GLMs, in the high-dimensional regime where the feature dimension $p$ and the sample size $n$ are both large and comparable. Precisely, we establish:

1. the limiting eigenvalue distribution of the Hessian matrix (Theorem 1); and
2. the behavior of (possible) isolated eigenvalue-eigenvector pairs (Theorem 2 and 3),

as a function of the dimension ratio $c = \lim p/n$, feature statistics, the loss function $\ell$ in (3), and the underlying response model in (1). Our results are based on a technical result of independent interest:

3. a *deterministic equivalent* (Theorem 4) of the random *resolvent* $\mathbf{Q}(z) = (\mathbf{H} - z\mathbf{I}_p)^{-1}$, for $z \in \mathbb{C}$ not an eigenvalue of $\mathbf{H}$, of the generalized sample covariance:[4]

$$\mathbf{H} \equiv \mathbf{H}(\mathbf{w}) = \tfrac{1}{n}\sum_{i=1}^{n} \ell''(y_i, \mathbf{w}^\mathsf{T}\mathbf{x}_i)\mathbf{x}_i\mathbf{x}_i^\mathsf{T} \equiv \tfrac{1}{n}\mathbf{X}\mathbf{D}\mathbf{X}^\mathsf{T}, \tag{4}$$

for $\mathbf{X} = [\mathbf{x}_1, \ldots, \mathbf{x}_n] \in \mathbb{R}^{p \times n}$, $\mathbf{D} \equiv \mathrm{diag}\{\ell''(y_i, \mathbf{w}^\mathsf{T}\mathbf{x}_i)\}_{i=1}^{n} \in \mathbb{R}^{n \times n}$, and $\ell''(y, h) \equiv \partial^2\ell(y, h)/\partial h^2$, as $n, p \to \infty$ with $p/n \to c \in (0, \infty)$, under the setting of *generic* Gaussian feature $\mathbf{x}_i \sim \mathcal{N}(\boldsymbol{\mu}, \mathbf{C})$, for $\boldsymbol{\mu} \in \mathbb{R}^p$ and positive definite covariance $\mathbf{C} \in \mathbb{R}^{p \times p}$. We also demonstrate our results empirically by showing that:

4. for a given response model (1), the Hessian eigenvalue distribution depends on the choice of loss function and the data/feature statistics in an intrinsic manner, e.g., bounded versus unbounded support and single- versus multi-bulk in Fig 2; and
5. there may exist two *qualitatively* different spikes—one due to data *signal* $\boldsymbol{\mu}$ and the other due to $\mathbf{w}_*$ or $\mathbf{w}$ and thus the *underlying model*—which may appear on different sides of the main bulk, and their associated phase transition behaviors are characterized (Fig 4 versus 5).

To have a more clear picture of our contribution, we compare, in Fig 1a and 1b, the Hessian eigenvalues for the logistic model (2) with the logistic loss $\ell(y, h) = \ln(1 + e^{-yh})$ (which is the loss of choice within the maximum likelihood framework), for different choices of $\mathbf{w}$ in the parameter space. A nontrivial interplay between the response model, feature statistics and the parameter $\mathbf{w}$ is reflected by the range of the Hessian eigenvalue support and an additional right-hand spike in Fig 1b, as confirmed by our theory. For phase retrieval model $y = (\mathbf{w}_*^\mathsf{T}\mathbf{x})^2$ with square loss $\ell(y, h) = (y - h^2)^2/4$, the non-convex nature of the problem is reflected by a (relatively large) fraction of negative Hessian eigenvalues in Fig 1c. We also note that the top eigenvector (that corresponds to the largest eigenvalue) contains structural information of the underlying model, in the sense that it is positively correlated with $\mathbf{w}_*$, as predicted by our theory. This is indeed connected to the Hessian-based initialization scheme widely used in non-convex problems [8, 37, 34, 40, 2, 51].

We conclude by emphasizing that, by focusing on the simple yet fundamental G-GLM, we obtain results that improve upon and are different than previous efforts in the following aspects:

(i) We provide *precise* asymptotic characterizations of the Hessian eigenspectra that go beyond, e.g., [7], where only Hessian lower bounds are given in the case of logistic model with logistic loss: our methodology and theoretical results hold much more generally for the family of G-GLM with arbitrary loss. As illustrating examples, we discuss linear least squares in Sec 3.1, logistic model with different choices of loss function in Fig 2, phase retrieval model in Figure 1c, and more in [43, Sec 4].

(ii) We extend the results in [57, 53, 44, 50] to G-GLMs by considering *generic* data statistics and loss function, whereas in previous efforts only much more homogeneous models are

---

[4]Here we consider $\mathbf{w}$ *independent* of $\mathbf{x}_i, y_i$. With some additional effort, our results extend to the case where $\mathbf{w}$ takes some simple *explicit* function form of $\mathbf{X}$ and $\mathbf{y}$, e.g., $\mathbf{w}$ is the least square solution to $(\mathbf{X}, \mathbf{y})$.

discussed, and sometimes under unrealistic assumption, e.g., the Hessian can be decomposed as the sum of two freely independent matrices, and the residual error, data feature, and weights are all composed of i.i.d. zero mean normal random variables [57, 58].

(iii) Instead of focusing solely on the main eigenvalue bulk as in [57, 58], our results also shed novel light on the isolated eigenvalues (above and/or below the bulk) that are empirically observed in the Hessian of modern NNs [60, 22, 42, 53, 52], as well as on the associated eigenvectors that are shown closely connected to NN training dynamics [28]. Also, relative to [57, 58], we show *qualitatively* different behaviors for the Hessian eigenspectra, e.g., bounded versus unbounded support, single- versus multi-bulk as in Figure 2. To our knowledge, these are *not* covered in the existing Hessian literature.

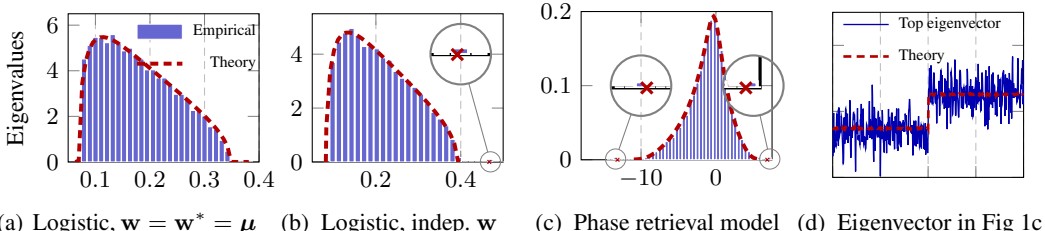

(a) Logistic, $\mathbf{w} = \mathbf{w}^* = \boldsymbol{\mu}$   (b) Logistic, indep. $\mathbf{w}$   (c) Phase retrieval model   (d) Eigenvector in Fig 1c

Figure 1: Illustration of our main results: eigenspectral properties of the Hessian of G-GLMs with $p = 800$, $n = 6\,000$ and $\mathbf{C} = \mathbf{I}_p$. **Fig 1a versus 1b**: absence versus presence of a right-hand side spike for different choices of $\mathbf{w}$, logistic model (2) with logistic loss, and $\mathbf{w}_* = \boldsymbol{\mu} \sim \mathcal{N}(\mathbf{0}, \mathbf{I}_p/p)$. **Fig 1c versus 1d**: the Hessian eigenspectra have a rather different shape (as opposed to the Marčenko-Pastur-like in Fig 1a and 1b) for the (non-convex) phase retrieval model (**1c**) and the top eigenvector is known in this case to be a (noisy) estimate of $\mathbf{w}_*$ (**1d**), as confirmed by our theory. With square loss $\ell(y, h) = (y - h^2)^2/4$, $\mathbf{w}_* = [-2 \cdot \mathbf{1}_{p/2}; \ 2 \cdot \mathbf{1}_{p/2}]/\sqrt{p}$, $\mathbf{w} \sim \mathcal{N}(\mathbf{0}, \mathbf{I}_p/p)$ and $\boldsymbol{\mu} = \mathbf{0}$.

## 1.3   Related work

Here, we provide a brief review of related previous efforts, see more discussions in [43].

**Random matrix theory.** Random matrices of the type (4) are related to the *separable covariance model* [81, 17] in the RMT literature, which is of the form $\mathbf{C}^{\frac{1}{2}}\mathbf{Z}\mathbf{D}\mathbf{Z}^\mathsf{T}\mathbf{C}^{\frac{1}{2}}$, for $\mathbf{Z}$ random and $\mathbf{C}, \mathbf{D}$ deterministic or random but *independent* of $\mathbf{Z}$. Our results generalize this, by allowing $\mathbf{D}$ to *depend* on $\mathbf{Z}$, in a possibly nonlinear fashion, per (4). This is of direct interest for the Hessian of G-GLMs.

**Hessian eigenspectra.** The eigenspectra of Hessian matrices arising in ML models (in particular, for NNs) have attracted considerable interest recently [60, 61, 11, 22, 72, 24, 32, 20, 64, 79, 80, 25]. However, these investigations are often built upon somewhat unrealistic simplifying assumptions and reduce to the "mixed" behavior of Marčenko–Pastur and semicircle law [57, 15]. In contrast, here we focus on the more tractable example of G-GLM and provide *precise* results on the Hessian eigenspectra for structural feature on arbitrary loss. Efforts have also been devoted to efficiently computing the eigenspectra of Hessian matrices, which, despite not enabling analytic results, allows for the computation or approximation of Hessian spectra for large-scale ML models such as DNNs, without any assumption on the data or underlying (response) model [30, 1, 79, 74, 25].

**Spectral initialization in non-convex problems.** A popular initialization scheme (of gradient-based methods) for non-convex problems is the *spectral initialization*, where the top eigenvectors of some Hessian-type matrices are used as gradient descent initialization [8, 37, 34, 40, 2, 51]. In [44], which was generalized in [50], the authors evaluated the eigenspectrum asymptotics of $\frac{1}{n}\sum_{i=1}^{n} f(y_i)\mathbf{x}_i\mathbf{x}_i^\mathsf{T}$, for $\mathbf{x}_i \sim \mathcal{N}(\mathbf{0}, \mathbf{I}_p)$. Their technical approach is, however, limited to the case of very homogeneous features with $\mathbf{x}_i \sim \mathcal{N}(\mathbf{0}, \mathbf{I}_p)$. Here we generalize the analysis in [44, 50] to the Hessian of G-GLM, by developing a systematic approach to account for both feature structures and loss functions.

**Scalable second-order methods.** Second-order methods are among the most powerful optimization methods that have been designed, and there have been several attempts to use their many advantages for machine learning applications [75, 69, 59], particularly for training NNs [79, 80, 64, 20, 73]. We

expect that our precise characterization of the Hessian sheds new light on the understanding and improved design of (e.g., computationally) more efficient second-order methods.

**Spectra of realistic NN weight matrices.** Recent work demonstrated that the eigenvalue distribution of weight matrices of state-of-the-art DNN models in computer vision and natural language applications exhibit heavy-tailed (i.e., *not* Marčenko–Pastur) behavior, and that this can be used to define metrics to predict trends in the quality of state-of-the-art NNs without access to training or testing data [47, 48, 46]. It would be of interest to extend our approach to weight matrix analysis.

## 2 Main results

In the section, we present our main results: on the limiting Hessian eigenspectrum (in Sec 2.1); and on the behavior of the (possible) isolated eigenvalue-eigenvector(s) (in Sec 2.2). These two main results depend on a technical deterministic equivalent result for the Hessian resolvent (in Sec 2.3), which is of independent interest. We position ourselves in the following high-dimensional regime.

**Assumption 1** (High-dimensional asymptotics). *As $n, p \to \infty$ with $p/n \to c \in (0, \infty)$, we have* $\max\{\|\mathbf{w}\|, \|\mathbf{w}_*\|\} = O(1)$ *and* $\mathbf{x}_i \overset{i.i.d.}{\sim} \mathcal{N}(\boldsymbol{\mu}, \mathbf{C})$ *with* $\max\{\|\boldsymbol{\mu}\|, \|\mathbf{C}\|\} = O(1)$.

### 2.1 Limiting spectral measure

Our first result is the limiting Hessian eigenvalue distribution. This is a direct consequence of our main technical Theorem 4 and is proven in [43, Sec A.2].

**Theorem 1** (Limiting spectral measure). *Let Assumption 1 hold, we have, as $n, p \to \infty$ with $p/n \to c \in (0, \infty)$, the empirical spectral measure[5] $\mu_{\mathbf{H}}$ of the Hessian matrix $\mathbf{H}$ defined in* (4) *converges weakly and almost surely to a probability measure $\mu$, defined through its Stieltjes transform $m(z) = \int (t - z)^{-1} \mu(dt)$ as the unique solution to[6]*

$$m(z) = \frac{1}{p} \operatorname{tr} \bar{\mathbf{Q}}_b(z), \quad \delta(z) = \frac{1}{n} \operatorname{tr} \left( \mathbf{C} \bar{\mathbf{Q}}_b(z) \right), \quad \bar{\mathbf{Q}}_b^{-1}(z) \equiv \mathbb{E}\left[ \frac{g \cdot \mathbf{C}}{1 + g \cdot \delta(z)} \right] - z \mathbf{I}_p, \quad (5)$$

*where*

$$g \equiv \partial^2 \ell(y, h)/\partial h^2, \quad for \ h = \mathbf{w}^\mathsf{T} \mathbf{x} \sim \mathcal{N}(\mathbf{w}^\mathsf{T} \boldsymbol{\mu}, \mathbf{w}^\mathsf{T} \mathbf{C} \mathbf{w}), \quad (6)$$

*and $y$ and $\ell$ defined respectively in* (1) *and* (3). *Moreover, if we denote $\nu$ the law/distribution of $g$ and assume the empirical spectral measure of $\mathbf{C}$ converges to $\tilde{\nu}$ as $p \to \infty$, then* (5) *writes*

$$m(z) = \int \left( -z + \tilde{t} \int \frac{t}{1 + t\delta(z)} \nu(dt) \right)^{-1} \tilde{\nu}(d\tilde{t}), \quad \delta(z) = \int \frac{c\tilde{t}}{-z + \tilde{t} \int \frac{t}{1 + t\delta(z)} \nu(dt)} \tilde{\nu}(d\tilde{t}). \quad (7)$$

In the form of (7), the (Stieltjes transform of the) limiting Hessian spectral measure $\mu$ is determined by the ratio $c = \lim p/n$ and the two measures $\nu$ and $\tilde{\nu}$. This formulation is closely connected to the separable covariance model [38, 6, 55, 17, 76] in RMT. Moreover, if $\nu(dt) = \delta_1(t)$ is a Dirac mass at one, this reduces to the popular sample covariance model [65]; taking further $\tilde{\nu}(dt) = \delta_1(t)$ gives the Marčenko-Pastur law. See Sec 3.1 for numerical evaluations of these special cases. In particular, the support of the (limiting) Hessian spectrum $\mu$ is directly linked to that of $\nu$ and $\tilde{\nu}$.

**Remark 1** (Hessian eigen-support). *Under Assumption 1, the (limiting) spectral measure $\tilde{\nu}$ of $\mathbf{C}$ has bounded support. However, this may not be the case for $\nu$, the law of $g$ defined in* (6). *Since the Hessian eigenvalue distribution $\mu$ is of compact support* if and only if *both $\nu$ and $\tilde{\nu}$ have compact support [17, Porposition 3.4], $\mu$ may be of unbounded support, depending on the model and the loss.*

An example of unbounded $\mu$ is the phase retrieval model with $y = (\mathbf{w}_*^\mathsf{T} \mathbf{x})^2$ and square loss $\ell(y, h) = (y - h^2)^2/4$, for which we have $g = 3(\mathbf{w}^\mathsf{T} \mathbf{x})^2 - (\mathbf{w}_*^\mathsf{T} \mathbf{x})^2$ for $\mathbf{x} \sim \mathcal{N}(\boldsymbol{\mu}, \mathbf{C})$. As a consequence, with say $\mathbf{w}_* = \mathbf{w}$, $g$ follows a chi-square distribution with one degree of freedom and has thus *unbounded* support. This corresponds to Fig 1c, where the Hessian spectrum has a "heavier" tail compared to Fig 1a (logistic model), and the empirically observed "isolated" eigenvalue is due to

---

[5]That is, the normalized counting measure of the eigenvalues $\lambda_i(\mathbf{H})$ of $\mathbf{H}$, i.e., $\mu_{\mathbf{H}} \equiv \frac{1}{p} \sum_{i=1}^{p} \delta_{\lambda_i(\mathbf{H})}$.

[6]Uniqueness is ensured in such a way that $\Im[m(z)] \cdot \Im[z] > 0$ for $\Im[z] \neq 0$ and $zm(z) < 0$ for $\Im[z] = 0$, so that $(z, m(z))$ is a valid Stieltjes transform couple, see more details in [29].

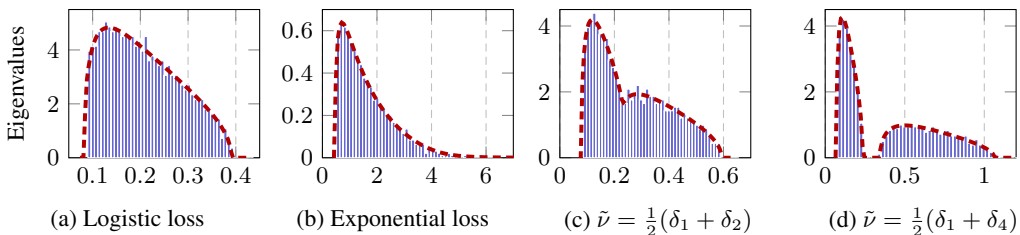

Figure 2: Impact of loss function: bounded (**2a**, with logistic loss) versus unbounded (**2b**, with exponential loss) Hessian eigenvalues, with $p = 800$, $n = 6\,000$, logistic model in (2) with $\boldsymbol{\mu} = \mathbf{0}$, $\mathbf{C} = \mathbf{I}_p$, $\mathbf{w}_* = \mathbf{0}$ and $\mathbf{w} = [-\mathbf{1}_{p/2},\ \mathbf{1}_{p/2}]/\sqrt{p}$. Impact of feature covariance: Hessian spectrum of single- (**2c**, with $\mathbf{C} = \mathrm{diag}[\mathbf{1}_{p/2};\ 2 \cdot \mathbf{1}_{p/2}]$) versus multi-bulk (**2d**, with $\mathbf{C} = \mathrm{diag}[\mathbf{1}_{p/2};\ 4 \cdot \mathbf{1}_{p/2}]$), with $p = 800$, $n = 6\,000$, logistic model with $\mathbf{w}^* = \mathbf{0}_p$, $\mathbf{w} = \boldsymbol{\mu} \sim \mathcal{N}(\mathbf{0}, \mathbf{I}_p/p)$.

a finite-dimensional effect and will be "buried" in the noisy main bulk for larger values of $n, p$. Therefore, aiming for an (almost surely) isolated eigenvalue-eigenvector (e.g., to recover the model parameter $\mathbf{w}_*$ using the top Hessian eigenvector), some pre-processing function $f$ must be applied. This has been discussed in previous work [44, 50] and corresponds to the so-called trimming strategy in phase retrieval [12], with for instance the truncation function $f(t) = \delta_{|t| \leq \epsilon}$ for some $\epsilon > 0$.

Another example of unbounded $\mu$ is when the exponential loss [23] is used. Precisely, consider the logistic model (2) with $\ell(y, h) = \exp(-yh)$, we have that $g = \exp(-yh)$ for $h \sim \mathcal{N}(\mathbf{w}^\mathsf{T}\boldsymbol{\mu}, \mathbf{w}^\mathsf{T}\mathbf{C}\mathbf{w})$ which follows a log-normal distribution and has unbounded support. As such, the (limiting) Hessian eigenvalue distribution $\mu$ has also unbounded support. On the other hand, with logistic loss $\ell(y, h) = \ln(1 + e^{-yh})$, one has $g \leq 1/4$ and $\mu$ is guaranteed to have bounded support. In Fig 2a and 2b, the empirical Hessian eigenvalues and the limiting distributions are compared for logistic and exponential losses, with a more "heavy-tailed" behavior observed for the exponential loss.

Clearly, depending on the measures $\nu$ (of $g$, which depends on feature statistics, loss and underlying model) and $\tilde{\nu}$ (of spectrum of feature covariance $\mathbf{C}$), the Hessian spectrum can have very different forms. Here we compare the empirical Hessian eigenvalues with their limiting behaviors per Theorem 1 for different feature covariance structures[7]. In particular, one may observe a single main bulk with more "compact" Hessian spectrum as in Fig 2c or multiple bulks (two in the case of Fig 2d) with Hessian eigenvalues more "spread-out", depending on the feature covariance structure $\tilde{\nu}$. In the form of (7), the condition for the existence of multi-bulk eigenspectrum has been thoroughly discussed in [17, Sec 3.2–3.4] and can be numerically evaluated with ease.

As a side remark, the "multi-bulk" behavior similar to Fig 2d has been empirically observed in Hessians of NNs in [42, 53] and is believed to be due to the classification structure within data (i.e., the data vectors are drawn from a mixture of distributions). Here, we provide an alternative explanation via feature covariance structure that holds beyond the classification setting.

## 2.2 Isolated eigenvalues and eigenvectors

As discussed in Remark 1, under Assumption 1, the Hessian has bounded (limiting) eigen-support if and only if $\nu$, the law of $g$, has bounded support. Under this condition (or, after the application of some function $f$ so that $f(g)$ is bounded), we can then talk about the (possible) isolated Hessian eigenvalues, as in the following result, the proof of which is given in [43, Sec A.3].

**Theorem 2** (Isolated eigenvalues). *In the setting of Theorem 1, assume that the law $\nu$ of the random variable $g$ defined in* (6) *is of bounded support, define*

$$\mathbf{G}(z) = \mathbf{I}_3 + \boldsymbol{\Lambda}(z)\mathbf{V}^\mathsf{T}\bar{\mathbf{Q}}_b(z)\mathbf{V} \in \mathbb{R}^{3\times3}, \tag{8}$$

*with $\bar{\mathbf{Q}}_b(z), \delta(z)$ defined in* (5)*, $\mathbf{V} \equiv [\boldsymbol{\mu},\ \mathbf{C}\mathbf{w}_*,\ \mathbf{C}\mathbf{w}] \in \mathbb{R}^{p\times3}$, $\mathbf{U} \equiv \mathbf{C}^{\frac{1}{2}}[\mathbf{w}_*,\ \mathbf{w}] \in \mathbb{R}^{p\times2}$ and*

$$\boldsymbol{\Lambda}(z) \equiv \mathbb{E}\frac{g}{1 + g \cdot \delta(z)}\begin{bmatrix} 1 & (\mathbf{U}^+\mathbf{z})^\mathsf{T} \\ \mathbf{U}^+\mathbf{z} & \mathbf{U}^+\mathbf{z}(\mathbf{U}^+\mathbf{z})^\mathsf{T} - (\mathbf{U}^\mathsf{T}\mathbf{U})^+ \end{bmatrix}, \quad \mathbf{z} = \mathbf{C}^{-\frac{1}{2}}(\mathbf{x}-\boldsymbol{\mu}) \sim \mathcal{N}(\mathbf{0}, \mathbf{I}_p), \tag{9}$$

---

[7]Covariance describes the joint variability or the "correlation" between entries of the feature vector and it is of particular significance in the analysis of image (with local structure) and time series data.

*where we denote $\mathbf{U}^+$ the Moore–Penrose pseudoinverse of $\mathbf{U}$. Then, for $\lambda$ such that $\mathbf{G}(\lambda)$ has a zero eigenvalue (of multiplicity one), there exists an eigenvalue $\hat{\lambda}$ of $\mathbf{H}$ such that $\hat{\lambda} - \lambda \xrightarrow{a.s.} 0$.*

Theorem 2 provides an asymptotic characterization of the possible isolated Hessian eigenvalues by computing the determinant of the much smaller (three-by-three) deterministic matrix $\mathbf{G}$ closely related to the key quantity $\delta(z)$ defined in Theorem 1. Note that, Theorem 2 does not provide, at least explicitly, the *phase transition* condition under which these spikes become "isolated" from the main bulk. As we shall see in more details in Sec 3.2, two types of quantitatively different phase transitions can be characterized, due to the data "signal" $\boldsymbol{\mu}$ and the underlying model, respectively.

We can also analyze the associated isolated eigenvectors. First note that, in the infinite data regime (i.e., for $n \to \infty$ with $p$ fixed), we have, by the strong law of large numbers, that $\mathbf{H} \xrightarrow{a.s.} \mathbb{E}[\mathbf{H}]$, with

$$\mathbb{E}[\mathbf{H}] = \mathbb{E}[\ell''(y, \mathbf{w}^\mathsf{T}\mathbf{x})\mathbf{x}\mathbf{x}^\mathsf{T}] = \mathbb{E}[g] \cdot \mathbf{C} + \mathbf{V} \begin{bmatrix} 1 & \mathbb{E}[g \cdot \mathbf{U}^+\mathbf{z}]^\mathsf{T} \\ \mathbb{E}[g \cdot \mathbf{U}^+\mathbf{z}] & \mathbf{U}^+\mathbb{E}[g \cdot (\mathbf{z}\mathbf{z}^\mathsf{T} - \mathbf{I}_p)](\mathbf{U}^+)^\mathsf{T} \end{bmatrix} \mathbf{V}^\mathsf{T}.$$

As a consequence, it is expected that in the large $n, p \to \infty$ limit, the top eigenvectors of $\mathbf{H}$ could also be related to the columns of $\mathbf{V}$. This is the case in Fig 1d, where the top eigenvector is observed to be a "noisy" version of the model parameter $\mathbf{w}_*$. More precisely, for $(\hat{\lambda}, \hat{\mathbf{u}})$ an isolated eigenpair of $\mathbf{H}$, the projection $\mathbf{V}^\mathsf{T}\hat{\mathbf{u}}\hat{\mathbf{u}}^\mathsf{T}\mathbf{V} \in \mathbb{R}^{3 \times 3}$ can be shown to be asymptotically close to a deterministic matrix. This measures the "cosine-similarly" between the Hessian isolated eigenvector $\hat{\mathbf{u}}$ with any column of $\mathbf{V}$ and consequently the performance of using $\hat{\mathbf{u}}$ as an estimate of, for instance the model parameter $\mathbf{w}_*$ for $\mathbf{C} = \mathbf{I}_p$.

This result is given in the following theorem, which is proven in [43, Sec A.4].

**Theorem 3** (Isolated eigenvectors). *In the setting of Theorem 2, for an isolated eigenvalue-eigenvector pair $(\hat{\lambda}, \hat{\mathbf{u}})$ of $\mathbf{H}$ and $\lambda$ the asymptotic position (of $\hat{\lambda}$) given in Theorem 2, then*

$$\mathbf{V}^\mathsf{T}\hat{\mathbf{u}}\hat{\mathbf{u}}^\mathsf{T}\mathbf{V} = -\mathbf{V}^\mathsf{T}\bar{\mathbf{Q}}_b(\lambda)\mathbf{V} \cdot \Xi(\lambda) + o(1), \quad \Xi(\lambda) = (\mathbf{v}_{l,\mathbf{G}}^\mathsf{T}\mathbf{G}'(\lambda)\mathbf{v}_{r,\mathbf{G}})^{-1} \cdot \mathbf{v}_{r,\mathbf{G}}\mathbf{v}_{l,\mathbf{G}}^\mathsf{T},$$

*for $\bar{\mathbf{Q}}_b(z)$ and $\mathbf{G}(z)$ defined in (5) and (8), respectively, $\mathbf{v}_{l,\mathbf{G}}, \mathbf{v}_{r,\mathbf{G}} \in \mathbb{R}^3$ the left and right eigenvectors of $\mathbf{G}(\lambda)$ associated with eigenvalue zero, and $\mathbf{G}'(\lambda)$ the derivative of $\mathbf{G}(z)$ with respect to $z$ evaluated at $z = \lambda$.*

## 2.3 Technical tool: deterministic equivalent

Our main technical tool to derive Theorem 1, 2 and 3 is a so-called deterministic equivalent [29, 16] result for the Hessian resolvent $\mathbf{Q}(z) = (\mathbf{H} - z\mathbf{I}_p)^{-1}$, that provides simultaneous access to the Hessian limiting eigenvalue distribution and the behavior of the possible isolated eigenpairs. Precisely, the normalized trace $\operatorname{tr}\mathbf{Q}(z)/p$ gives the Stieltjes transform $m_{\mathbf{H}}(z) = \int(t-z)^{-1}\mu_{\mathbf{H}}(dt)$ of the empirical spectral measure $\mu_{\mathbf{H}}$ of $\mathbf{H}$, from which $\mu_{\mathbf{H}}$ can be recovered. Also, for $(\hat{\lambda}, \hat{\mathbf{u}})$ an eigenpair of interest, with Cauchy's integral formula we have $|\mathbf{w}^\mathsf{T}\hat{\mathbf{u}}|^2 = -\frac{1}{2\pi i}\oint_{\Gamma_\lambda} \mathbf{w}^\mathsf{T}\mathbf{Q}(z)\mathbf{w}\,dz$, for a deterministic vector $\mathbf{w} \in \mathbb{R}^p$ and $\Gamma_\lambda$ a positively oriented contour surrounding *only* $\hat{\lambda}$. As such, for $\bar{\mathbf{Q}}(z)$ a deterministic equivalent of $\mathbf{Q}(z)$, that is, $\mathbf{Q}(z) \leftrightarrow \bar{\mathbf{Q}}(z)$ with $\operatorname{tr}\mathbf{A}(\mathbf{Q}(z) - \bar{\mathbf{Q}}(z))/p \to 0$ and $\mathbf{a}^\mathsf{T}(\mathbf{Q}(z) - \bar{\mathbf{Q}}(z))\mathbf{b} \to 0$ almost surely as $n, p \to \infty$, for $\mathbf{A} \in \mathbb{R}^{p \times p}$ and $\mathbf{a}, \mathbf{b} \in \mathbb{R}^p$ of bounded (Euclidean and spectral) norms, the limiting spectral measure (via the associated Stieltjes transform) and the isolated eigenpairs of $\mathbf{H}$ are directly accessible via the study of the deterministic equivalent $\bar{\mathbf{Q}}(z)$. This result is given as follows, with the proof deferred to [43, Sec A.1].

**Theorem 4** (Deterministic equivalent). *Let $\mathbf{Q}(z) \equiv (\mathbf{H} - z\mathbf{I}_p)^{-1}$ be the resolvent of $\mathbf{H}$ defined in (4). Then, under Assumption 1, as $n, p \to \infty$ with $p/n \to c \in (0, \infty)$,*

$$\mathbf{Q}(z) \leftrightarrow \bar{\mathbf{Q}}(z), \quad with \ \bar{\mathbf{Q}}^{-1}(z) = \mathbb{E}\left[\frac{g}{1 + g \cdot \delta(z)}(\mathbf{C}^{\frac{1}{2}}(\mathbf{I}_p - \mathbf{P}_\mathbf{U})\mathbf{C}^{\frac{1}{2}} + \boldsymbol{\alpha}\boldsymbol{\alpha}^\mathsf{T})\right] - z\mathbf{I}_p,$$

*for random vector $\boldsymbol{\alpha} \equiv \boldsymbol{\mu} + \mathbf{C}^{\frac{1}{2}}\mathbf{P}_\mathbf{U}\mathbf{z} \in \mathbb{R}^p$ and $g = \ell''(y, \mathbf{w}^\mathsf{T}\boldsymbol{\mu} + \mathbf{w}^\mathsf{T}\mathbf{C}^{\frac{1}{2}}\mathbf{z})$ for $\mathbf{z} \sim \mathcal{N}(\mathbf{0}, \mathbf{I}_p)$ as defined in (6), $y$ and $\delta(z)$ defined in (1) and (5), respectively, and $\mathbf{P}_\mathbf{U} \in \mathbb{R}^{p \times p}$ the projection onto the subspace spanned by the columns of $\mathbf{U} \equiv \mathbf{C}^{\frac{1}{2}}[\mathbf{w}_*, \ \mathbf{w}]$.*

## 3 Evaluations and Discussions

In this section, we provide further discussions on the consequences of Theorem 1, 2 and 3, together with numerical evaluations. Implications of Theorem 1 on the Hessian eigenvalue distribution is

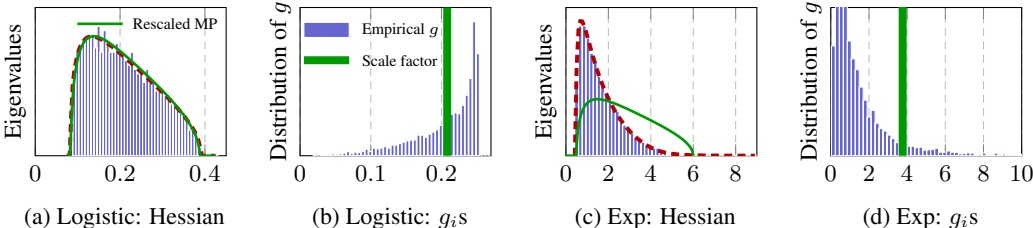

| (a) Logistic: Hessian | (b) Logistic: $g_i$s | (c) Exp: Hessian | (d) Exp: $g_i$s |

Figure 3: Empirical Hessian eigenspectra (**blue**, with limiting behavior in **red** per Theorem 1) versus rescaled and shifted Marčenko-Pastur laws (**green**) in the setting of Fig 2 with $p = 800$ and $n = 6\,000$. **Fig 3a versus 3b**: Marčenko-Pastur-like Hessian with logistic loss, Hessian eigenvalues (**3a**) and empirical distribution of the $g_i$s versus the scaling factor (**green** in **3b**, obtained by empirically matching the minimal and maximal empirical Hessian eigenvalues to the Marčenko-Pastur law). **Fig 3c versus 3d**: an example of non-Marčenko-Pastur-like Hessian with exponential loss and the associated $g_i$s (**green**). Note that the scales of the axes are different in different subfigures.

discussed in Sec 3.1. In Sec 3.2, we discuss the consequences of Theorem 2 and 3 on the possible isolated eigenpairs, for which two fundamentally different phase transitions are characterized.

## 3.1 Hessian eigenvalues distribution

For a better interpretation of Theorem 1 on the Hessian eigenspectrum, we consider here the special case of $\mathbf{C} = \mathbf{I}_p$, and start with the simple setting where the random variable $g$ in (6) is constant, say $g = 1$: this happens, e.g., when the square loss $\ell(y, h) = (y - h)^2/2$ is employed. In this case, the Hessian does *not* dependent on $\mathbf{w}, \mathbf{w}_*$ and the Stieltjes transform $m(z)$ is the solution to $zcm^2(z) - (1 - c - z)m(z) + 1 = 0$ and corresponds to the Marčenko-Pastur law.

As long as $g$ is *not* constant, the limiting Hessian spectrum is, a priori, different from the Marčenko-Pastur law, even in the $\mathbf{C} = \mathbf{I}_p$ setting, since the associated Stieltjes transform $m(z)$ is different from the solution to the Marčenko-Pastur equation. However, we see in Fig 3a that, for the logistic model (2) with logistic loss, the Hessian spectrum is close, at least visually, to a (rescaled) Marčenko-Pastur law. This can be understood with Theorem 1 and is due to the fact that, the distribution of $g$ is more "concentrated" (around some constant, see Fig 3b versus 3d for a comparison between different cases). This is in sharp contrast to Fig 3c where with the exponential loss, the law of $g$ has a much larger spread and the Hessian is therefore away from a Marčenko-Pastur-shape.

This "empirical fit" has been observed in [56, Fig 5], where acceleration methods proposed for a Marčenko-Pastur distributed Hessian (in linear least squares) work reasonably well on logistic regression models. Our theory proposes a convincing theoretical explanation of this empirical observation on logistic regression, and possibly for others more involved ML models. Nonetheless, it must be pointed out that this "visual approximation" by Marčenko-Pastur law is *not robust*, in the sense that it "visually" holds only for, yet formally different from, the case of (i) logistic model with (ii) logistic loss and (iii) identity covariance $\mathbf{C} = \mathbf{I}_p$: any change in the response model (e.g., the phase retrieval model in Fig 1c), in the choice of loss function (e.g., the exponential loss in Fig 2b), or beyond the identity covariance setting (as in Fig 2c and 2d) would induce a Hessian spectrum that is very different from the Marčenko-Pastur law. In this vein, our Theorem 1 goes beyond such "loose" Marčenko-Pastur approximation and acts as a more accurate first example in the understanding of Hessian in more involved ML models beyond linear least squares that accounts for nonlinear transformations (such as activation function in NNs) and feature statistics.

While Theorem 1 is proven here only for Gaussian features, we conjecture, as is the case for many random matrix asymptotics, that it holds more generally beyond Gaussian distribution, see [43, Fig 5] and more discussions therein.

## 3.2 Isolated eigenvalues and their phase transitions

In this section, we discuss the implications of Theorem 2 and 3 on the possible isolated eigenvalue-eigenvector pairs. More precisely, we show that, different from the classical spiked models extensively studied in RMT literature [3, 5, 41], for which (i) the isolated spike appears due to the presence

of some statistical "signal" in the data and (ii) a "monotonic" phase transition behavior can be characterized as a function of the signal strength; here another type of Hessian spike arises due to the underlying G-GLM model (i.e., $\mathbf{w}_*$ and $\mathbf{w}$) and exhibits a rather different behavior.

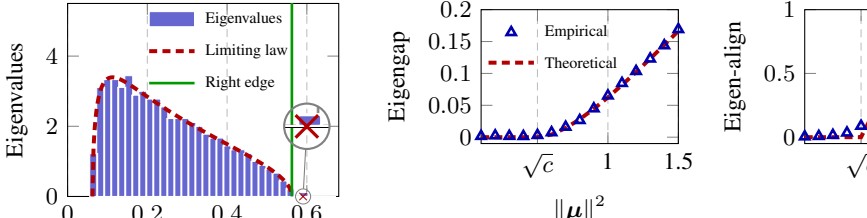

Figure 4: Spike due to data signal in Corollary 1: theory versus practice of **(left)** Hessian eigen-spectrum with $\|\boldsymbol{\mu}\|^2 = 0.8$, **(middle)** eigengap $\mathrm{dist}(\lambda_{\boldsymbol{\mu}}, \mathrm{supp}(\mu))$, and **(right)** top eigenvector alignment $\alpha$ in (10), as a function of the signal strength $\|\boldsymbol{\mu}\|^2$, on logistic model with logistic loss, for $\boldsymbol{\mu} \propto [-\mathbf{1}_{p/2}, \ \mathbf{1}_{p/2}]$, $\mathbf{w} = \mathbf{w}_* = \mathbf{0}$, $\mathbf{C} = \mathbf{I}_p$, $p = 512$ and $n = 2\,048$. Results averaged over 50 runs.

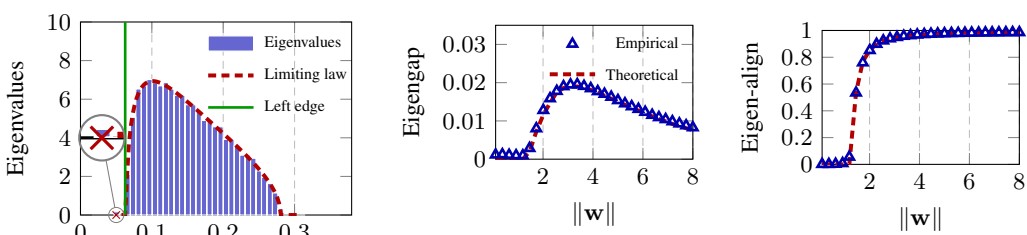

Figure 5: Left-hand side spike due to response model in Corollary 2 in the *absence of data signal*: **(left)** Hessian spectrum for $\|\mathbf{w}\| = 2$, with a *left* isolated eigenvalue $\hat{\lambda}_{\mathbf{w}}$, **(middle)** eigengap $\mathrm{dist}(\lambda_{\mathbf{w}}, \mathrm{supp}(\mu))$, and **(right)** dominant eigenvector alignment (with $\mathbf{w}$), as a function $\|\mathbf{w}\|$ with $\mathbf{w} \propto [-\mathbf{1}_{p/2}, \ \mathbf{1}_{p/2}]$, $\mathbf{w}_* = \boldsymbol{\mu} = \mathbf{0}$, $\mathbf{C} = \mathbf{I}_p$, $p = 800$ and $n = 8\,000$. Results averaged over 50 runs.

### 3.2.1 Spike due to data signal

To study the spike due to data "signal" $\boldsymbol{\mu}$ and its phase transition behavior, we focus here on the case $\mathbf{w}_* = \mathbf{w} = \mathbf{0}$. This, in the case of logistic model (2) for example, gives rise to a much simpler form of limiting spectrum (per Theorem 1) and possible isolated eigenpairs (per Theorem 2 and 3), as summarized in the following corollary, with detailed derivations given in [43, Sec B.3].

**Corollary 1** (Spike due to data signal: logistic model)**.** *Consider the logistic model in* (2) *with logistic loss, for* $\mathbf{w} = \mathbf{w}_* = \mathbf{0}$ *and* $\mathbf{C} = \mathbf{I}_p$, *the limiting Hessian eigenvalue distribution is the Marčenko-Pastur law, but rescaled by a factor of* $g = 1/4$. *Moreover, there is* at most one *isolated eigenpair* $(\hat{\lambda}_{\boldsymbol{\mu}}, \hat{\mathbf{u}}_{\boldsymbol{\mu}})$ *of* $\mathbf{H}$ *and it satisfies*

$$\hat{\lambda}_{\boldsymbol{\mu}} \xrightarrow{a.s.} \begin{cases} \lambda_{\boldsymbol{\mu}} = \frac{1}{4}(1 + \rho + c \cdot \frac{\rho+1}{\rho}) & \rho > \sqrt{c}, \\ \frac{1}{4}(1 + \sqrt{c})^2 & \rho \le \sqrt{c}; \end{cases}, \quad \frac{|\boldsymbol{\mu}^{\mathsf{T}} \hat{\mathbf{u}}_{\boldsymbol{\mu}}|^2}{\|\boldsymbol{\mu}\|^2} \xrightarrow{a.s.} \begin{cases} \alpha = \frac{\rho^2 - c}{\rho^2 + c\rho} & \rho > \sqrt{c}, \\ 0 & \rho \le \sqrt{c}; \end{cases} \tag{10}$$

*with the signal strength* $\rho = \lim_{p \to \infty} \|\boldsymbol{\mu}\|^2$ *and* $c = \lim p/n$.

The behavior of the isolated eigen-pairs described in Corollary 1 follows the "classical" phase transition [4, 3, 54]: (i) the isolated eigenvalue always appears on the right-hand side of the main (Marčenko-Pastur) bulk and (ii) the eigenvalue amplitude and eigenvector alignment is "monotonic" with respect to the signal strength $\|\boldsymbol{\mu}\|^2$ in the sense that, for a fixed dimension ratio $c$, the largest Hessian eigenvalue is bound to become asymptotically isolated once $\|\boldsymbol{\mu}\|^2$ exceeds $\sqrt{c}$ and its value, as well as the eigenvector alignment, increase monotonically as $\|\boldsymbol{\mu}\|^2$ grows. This is confirmed in Fig 4. As we shall see below, this is *not* the case for, e.g., the spike due to model parameter $\mathbf{w}$.

### 3.2.2 Spike due to model

To investigate the spike due to the underlying model (i.e., $\mathbf{w}_*$ and $\mathbf{w}$), we position ourselves in the situation where $\boldsymbol{\mu} = \mathbf{0}$, that is, in the absence of data "signal". This leads to the following corollary, the proof of which is given in [43, Sec B.4].

**Corollary 2** (Spike due to model: logistic model). *Consider the logistic model in* (2) *with logistic loss, $\boldsymbol{\mu} = \mathbf{0}$ and $\mathbf{C} = \mathbf{I}_p$, then the Stieltjes transform $m(z)$ satisfies $m(z) = 1/(\mathbb{E}[f(r,z)] - z)$ for $f(r,z) = 1/(cm(z) + 2 + e^{-r} + e^r)$ and $r \sim \mathcal{N}(0, \|\mathbf{w}\|^2)$ that depends on $\mathbf{w}$ but not on $\mathbf{w}_*$. Moreover, there is at most one isolated eigenvalue $\hat{\lambda}_{\mathbf{w}}$ of $\mathbf{H}$ that is due to $\mathbf{w}$ and satisfies $\hat{\lambda}_{\mathbf{w}} - \lambda_{\mathbf{w}} \xrightarrow{a.s.} 0$ with $\lambda_{\mathbf{w}}$ solution to $0 = \det \mathbf{G}(\lambda_{\mathbf{w}}) = 1 + m(\lambda_{\mathbf{w}}) \frac{\mathbb{E}[f(r,\lambda_{\mathbf{w}})(r^2 - \|\mathbf{w}\|^2)]}{\|\mathbf{w}\|^2}$.*

The situation here is more subtle (than the spike due to data signal discussed in Sec 3.2.1): as the model parameter $\mathbf{w}$ changes (e.g., as the "energy" $\|\mathbf{w}\|$ grows), both the Hessian (limiting) eigenvalue distribution and the possible spike location are impacted. Fig 5 illustrates the behavior of the spike due to $\mathbf{w}$ in the setting of Corollary 2. Note first that, different from the case of spike due to data signal $\boldsymbol{\mu}$, the spike in Fig 5-(left) appears on the *left-hand side* of the main bulk: this particularly means that the Hessian may admit an eigenvalue that is *significantly smaller* than all the other eigenvalues.[8] Also, note from Fig 5-(middle) that, different from the spike due to $\boldsymbol{\mu}$, the spike due to $\mathbf{w}$ exhibits here a "non-monotonic" behavior in the sense that, it is absent for small values of $\|\mathbf{w}\|$ (as for small $\|\boldsymbol{\mu}\|$ in Fig 4-middle) and becomes "isolated" as $\|\mathbf{w}\|$ increases, but then again "merges into" the main bulk as $\|\mathbf{w}\|$ continues to increase, resulting an eigengap that falls back to zero.

It is perhaps even more surprising to observe in Fig 5-(right) that, the alignment between the associated isolated eigenvector and the parameter $\mathbf{w}$ is, unlike the eigengap in Fig 5-(middle), monotonically increasing as $\|\mathbf{w}\|$ grows large, as in the case of Fig 4-(right). This suggests that, in the case of spike due to model, a *smaller* eigengap may not always imply *less* statistical "information" contained in the associated eigenvector, which somehow goes against the conventional eigengap heuristic [67, 35]. It is worthy mentioning that, while, technically speaking, the proposed analysis is not capable of characterizing the behavior as $\|\mathbf{w}\| \to \infty$ under Assumption 1, empirical results suggest that for extremely large $\|\mathbf{w}\|$, the eigengap tends to "vanish", the associated dominant eigenvector can still be used to recover $\mathbf{w}$ almost perfectly, see [43, Fig 8] as an example.

## 4 Conclusion

In this article, we provided a precise asymptotic characterization of Hessian eigenspectra (including the limiting eigenvalue distribution and the behavior of possible isolated eigenvalue-eigenvector pairs) for the family of G-GLMs via a random matrix analysis. By assuming generic Gaussian input features, our results show that the Hessian can have a "heavy-tailed" or multi-bulk spectral behavior, and with isolated eigenvalues of the left- or right-hand side of the main bulk, depending on the underlying GLM and the choice of loss function.

Extending our approach to more involved ML models such as multi-layer neural networks is challenging, e.g., one will need to first study the layer-wise Hessian and evaluate how the objective of interest (e.g., the eigenvalue distribution) of the full Hessian depends on each of its (layer-wise) blocks. Simplification can be made by specifying the choice of loss function and/or using different approximations (such as Gauss-Newton-type approximation [63] or working in the infinitely-wide but less-realistic NTK regime [32]), so as to facilitate analysis. We leave this for future work.

## Acknowledgments and Disclosure of Funding

ZL would like to acknowledge the Fundamental Research Funds for the Central Universities of China (2021XXJS110) and CCF-Hikvision Open Fund (20210008) for providing partial support. MWM would like to acknowledge DARPA, IARPA (contract W911NF20C0035), NSF, and ONR via its BRC on RandNLA for providing partial support. Our conclusions do not necessarily reflect the position or the policy of our sponsors, and no official endorsement should be inferred.

---

[8]Depending on the response model and loss, the spike due to model may also appear on the right-hand side of the bulk or even establish a left-to-right transition. The eigenvector alignment can behave differently from Fig 4-5 and establish a non-monotonic behavior as a function of $\|\mathbf{w}\|$ or $\|\mathbf{w}_*\|$; see [43, Sec 4.3].

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
