# OpenReview forum: "Hessian Eigenspectra of More Realistic Nonlinear Models"
_NeurIPS.cc/2021/Conference — NeurIPS 2021 Oral_

### Official Review · Reviewer_d8k3 · 2021-06-29

**Rating:** 8
**Confidence:** 4

**Summary:**

The paper explores the spectrum of Hessians for a class of generalized generalized linear models which includes convex problems such as logistic loss and non-convex problems such as the phase retrieval. The authors make some simplifying assumptions such as data $x_i \sim N(\mu, C)$ so that random matrix theory tool can be applied. The result is a complete characterization of the spectrum of the Hessian via the Stieltjes transform which shows how the covariance in the data, the loss function, and the distribution of the feature vector each effect the density of the eigenvalues. In particular, they show that the Hessian $H(w)$ is deterministically equivalent to a generalized sample covariance matrix $H(w) = 1/n X D X^T$ where $D$ is a diagonal matrix with the second derivatives of the loss evaluated at the current $w$. The authors go on to analyze various properties of $H$ for logistic loss and phase retrieval in detail and show that for many cases the Hessian does not resemble a pure Marchenko-Pastur.

Some other contributions:

(1). They show that the spectrum of the Hessian $H$ is unbounded due to either (i) the ESD of the covariance in the data converging to an unbounded measure (this is known even for the classes spiked models) and (ii) if $g := \frac{\partial^2 \ell}{\partial h^2} |_{x^T w} \sim \nu$ is unbounded. In the case of phase retrieval, the $g$ follows a chi-distribution and therefore unbounded; this accounts for the heavy-tailed features in the spectrum.

(2). The authors show that there are two different reasons for an outlier eigenvalue to occur. The first is that the data itself is not normalized to be 0 mean and so the $\mu$ creates a spike (usually to the left of the bulk). The second outlier occurs because of the parameters $w$.

**Ethical Concerns:**

There are no ethical concerns; the result is purely theoretical.

**Limitations And Societal Impact:**

The work is purely theoretical. The authors made clear their assumptions (e.g. the data matrices distributed as Gaussian and the class of objective functions they are considering).

**Main Review:**

The paper extends the models for Hessians in novel and practical way to generalized generalized linear models (e.g. nonconvex objective functions such as phase retrieval) and adds covariance structure. The paper is well-written with some simulated data to show how their theoretical results match the empirical spectral distributions. Although I did not check all the details for the mathematics, I am confident that results are correct.

I have a few clarifying questions for the authors, but otherwise I am happy to accept this paper. It has some interesting random matrix theory results for Hessians.

(1). If the largest eigenvalue converges to the outlier, does the second largest eigenvalue converge to the edge of the bulk
(2). Could the authors clarify on the assumptions on $w$ versus $w_*$? For instance, is $w^*$ or $w$ chosen independent of $X$? More importantly how are the targets $y$ chosen and does this choice effect anything? Does adding noise to the targets effect anything?

**Time Spent Reviewing:**

3

---

> ### Author Response · Authors · 2021-08-06
> **Response to Reviewer d8k3**
>
> We would like to thank the reviewer for his/her positive support and for the thorough and helpful remarks. To clarify the reviewer's questions (and hopefully to even push the Accept recommendation to Strong Accept recommendation), here are the answers (which we will clarify in the final version of the submission):
> * Question: if the largest eigenvalue converges to the outlier, does the second largest eigenvalue converge to the edge of the bulk (2).
> * Answer: yes, this so-called "no eigenvalue outside the support" result for the separable covariance under study here is established in the paper "No eigenvalues outside the support of the limiting empirical spectral distribution of a separable covariance matrix", we will clarify this point in a revised version of the manuscript.
> * Question: Could the authors clarify the assumptions on $w$ versus $w_*$?
> * Answer: $w$ and $w_*$ are (i) either fixed with respect to the random $X$, or (ii) independent of the random $X$ (while the target $y$ depend on $w_*$). We will clarify this point in a revised version of the manuscript.
> * Question: More importantly how are the targets $y$ chosen and does this choice affect anything? Does adding noise to the targets affect anything?
> * Answer: the targets $y$s are independently drawn from Equation (1) and act on our main results in Theorem 1 via the (law of the) random variable $g$ defined in (6). In this respect, adding noise to the targets changes the law of $g$ and consequently the characterization in (7) of Theorem 1.

---

> > ### Comment · Reviewer_d8k3 · 2021-08-15
> > **Comments from the Response**
> >
> > Thank you so much for your response. I would advocate this for acceptance as well and I will update my score accordingly.

---

### Official Review · Reviewer_3BGa · 2021-07-07

**Rating:** 9
**Confidence:** 3

**Summary:**

This paper studies generalized linear models in the large dimension, large sample size limit, obtains an analytic expression for the Hessian of the loss function, and discusses many qualitative features such as the relation to random matrix results and the appearance of isolated eigenvalues.

**Limitations And Societal Impact:**

No apparent societal impact.

**Main Review:**

This paper studies a very broad class of generalized linear models including the standard ones, phase retrieval, and essentially any model with a response distribution depending on $w_* \cdot x$ and a loss function of the form $ \sum_{i\in\mbox{samples}} L(y_i,w\cdot x_i) $.  An analytic expression for the Hessian is obtained for the large dimension p and large sample size n limit with fixed ratio p/n.  One assumes that the data distribution is approximately normal in this limit.  The derivation is an explicit computation, more or less decomposing the Hessian along the  $w_*,w$ and orthogonal directions and thus obtaining a spiked covariance model.  The result has a similar analytic form to those obtained using random matrix theory and reduces to them in special cases, but takes into account the full nonlinearity of the loss function and the resulting spectra can be quite different.
Comparisons are made to studies of the Hessian for particular models: logistic, limits of neural networks, many empirical studies.
The qualitative features seen there, in particular isolated eigenvalues and "multi-bulk" spectra, are reproduced.
Previous observations that empirical results are not fit by RMT results such as Marcenko-Pastur are explained.
A new isolated eigenvalue (or "spike") whose location is a function of the model parameter w is sometimes present.

This looks like quite a general and powerful result and I am enthusiastic about the paper.  Although similar techniques are used in many recent papers, so far as I know the result in the generality stated here is new.

**Time Spent Reviewing:**

1.5

---

> ### Author Response · Authors · 2021-08-06
> **Response to Reviewer 3BGa**
>
> We thank the reviewer for the positive support.

---

### Official Review · Reviewer_MLx1 · 2021-07-11

**Rating:** 8
**Confidence:** 3

**Summary:**

This papers develops new results in random matrix theory applied to characterize the limiting spectral distribution of Hessian matrices of common loss functions.  Specifically,  the setting is what the authors call 'generalized GLMs',  i.e. a class of models that extends classical generalized linear models.   The paper addresses a shortcoming in the literature where much of the analysis of Hessian matrices uses fairly strong simplifying assumptions (e.g. such as Marcenko-Pastur).  Even though the paper doesn't consider deep NNs,  even within the context of G-GLMs it finds rich predictable spectral structure not yet accounted for in the literature, and should have important consequences.

**Ethical Concerns:**

N/A.

**Limitations And Societal Impact:**

N/A.

**Main Review:**

The main contribution of the paper is a new mathematical result describing the empirical spectral measure of Hessian matrices for G-GLMs.  The result employs random matrix theory (RMT) tools: Stieltjes transforms, deterministic equivalents of random resolvents, free probability, e.t.c.), and extends existing results for so-called 'separable covariance models' to capture G-GLMs in full generality.  The RMT line of attack has been considered in earlier papers e.g. Benaych-Georges and Nadakuduti [4],  and Pennington and Worah [52]. However, the simplifying assumptions used in these papers to make the analysis more mathematically tractable lose important detail of the model.  This paper goes through the derivations not making these assumptions.  Similar to Marcenko-Pastur it provides remarkably accurate predictions of  empirical spectral distributions, and is able to describe important salient features of such spectra (existence of isolated eigen-values,  the shape of the bulk of the distribution,  whether the distribution is a single-bulk or multi-bulk (multi-modal),  and whether it has finite or infinite support).

I believe that the paper offers the ML community a new powerful tool which would be valuable in getting new more accurate insight into both G-GLMs,  and (eventually) deep NNs, and perhaps help pave the way for more efficient second-order SGD algorithms.  The paper is well written, and shows important implications of their results (from initialization of non-convex models, to inadequacy of currently popular assumptions on Hessian analysis).  I would have liked to see some discussion / hints on how to extend these results to NNs where they'd be of most interest,  and some comments on large-scale approximate spectral computations.  Furthermore, since RMT tools may be out of reach for many ML researchers, I would recommend to put together a toolbox that can predict/estimate Hessian spectra for moderate-size models.

Comments:
1) The term 'generalized generalized linear model'  sounds a bit awkward (I thought it's a typo at first).  I don't have great suggestions here -- but since it's a natural generalization of GLMs -- it's worth thinking about a better name.  E.g. Extended GLMs,  or wide-sense GLMs...

2) Since you extend the RMT analysis from papers such as [52] which ultimately aim to analyze deep NNs,  it may be worth summarizing a pathway of how to use your results for multi-layer NNs.

3) It's worth adding more context for computational tools from randomized-linear algebra that can be used to compute approximate eigen-spectra of Hessian matrices for very large models.  You cite [72] among many citations (there are also related jax tools), but provide little context -- e.g. while they do not enable analytic results, they are indeed able to compute accurate Hessian spectral densities for large deep networks without any assumptions and can be used for analysis of specific models.
* Adams, Ryan P., et al. "Estimating the spectral density of large implicit matrices." arXiv preprint arXiv:1802.03451 (2018).
* Han, Insu, et al. "Approximating spectral sums of large-scale matrices using stochastic chebyshev approximations." SIAM Journal on Scientific Computing 39.4 (2017): A1558-A1585.

4) As you're working in the high-dimensional setting (n,p large), Hessian spectra can be thought of as being 'smoothed-out' due to finite-sample noise.  Could there be value in applying 'deconvolution' tools as proposed in
El Karoui, Noureddine. "Spectrum estimation for large dimensional covariance matrices using random matrix theory." The Annals of Statistics 36.6 (2008): 2757-2790.
to the G-GLM setting?

5) Minor:  you use loss and objective in the first few sentences -- which is a bit confusing (e.g. perhaps objective could involve regularization terms in addition to loss).  Worth using consistent language.

6) Are there any success stories in second-order SGD in practice?  It's worth mentioning if this is still an active research area, or if there are indeed practical existing algorithms.  E.g. LBFGS is not exactly a Hessian-based method.

7) page 1: typo faction -> fraction.

8) page 3:  "(maximum likelihood) logistic loss".  Unclear -- do you mean Hessian at the ML-estimate of the logistic loss?  Then it doesn't make sense to compute it for 'different choices of w in the parameter space'.  Rewrite to make it clearer.

9) It's worth adding a reference / definition for "freely independent matrices".  Also not clear what you mean by 'very homogeneous features'.  Also the term 'the law of g' can be unclear.

10) For random Z and C, D independent of Z.  It's unclear what's random -- if D is not random -- then what does it mean to be 'independent' of Z?

11) Is the setting for figure 3 again p=800, n=6000?  The meaning of the green bar in (d) is confusing.

12)  Page 9:  are isolated eigenvalues above the bulk and below the bulk equally interesting?

**Time Spent Reviewing:**

4

---

> ### Author Response · Authors · 2021-08-06
> **Response to Reviewer MLx1**
>
> We would like to thank the reviewer for his/her positive support and for the thorough and helpful remarks. In an updated version of the submission, we will add (i) more discussions on the possible extension to the NN context, (ii) references and discussions on large-scale approximate spectral computation (of NN Hessians particularly) mentioned by the reviewer, (iii) fix typos and add clarifications pointed out by the reviewer.
>
> * Question: The term 'generalized generalized linear model' sounds a bit awkward (I thought it's a typo at first). I don't have great suggestions here -- but since it's a natural generalization of GLMs -- it's worth thinking about a better name. E.g. Extended GLMs, or wide-sense GLMs...
> * Answer: we agree that the term is a little awkward, but we haven't found a better alternative; we'll keep trying before the final version is due, since we definitely don't want confusion.
> * Question: page 3: "(maximum likelihood) logistic loss". Unclear
> * Answer: here we refer to the logistic loss, which is known to be the choice of loss in the maximum likelihood (ML) framework for the logistic model, and we do not specify the choice of parameter $w$ (to e.g., the ML estimate). We will clarify this point in a revised version of the manuscript.
> * Question: Also not clear what you mean by 'very homogeneous features'.
> * Answer: we refer to the case of Gaussian data/features with zero mean and identity covariance, which is a special case of the data model $x \sim \mathcal N(\mu, C)$ under study here. We will clarify this point in a revised version of the manuscript.

---

> > ### Comment · Reviewer_MLx1 · 2021-08-22
> > **response to authors**
> >
> > Thank you for the explanations, this is helpful! Nice work!

---

### Official Review · Reviewer_rM2S · 2021-07-16

**Rating:** 9
**Confidence:** 4

**Summary:**

This paper studies the spectrum of empirical cost functions used with data from generalized generalized linear models (G-GLM). Such models depend on some parameters $w^* \in \mathbb{R}^p$ that link the input features $x_i \in \mathbb{R}^p$ to the responses $y_i \in \mathbb{R}$ by the relation $y_i \sim  f(y |  (w^*)^T x_i)$
where $f(\cdot | \cdot)$ is some conditional density function.
Given a training set $(x_i, y_i)$ for $i=1, \dots, n$ a popular approach to estimate $w_*$ is to minimize the cost function
$
L(w) = \sum_{i=1}^n \ell(y_i, x_i^T w)
$
for some loss $\ell$ that is carefully chosen.

This framework is quite general and encompass many classical models such as the standard linear model, the logistic model or the phase retrieval problem.


The authors studies this model in the setting where
- $n,p \to \infty$, while $n/p$ converges to some positive number.
- $x_i \sim_{iid} \mathcal{N}(\mu, C)$
- $\|w\|, \|w_*\|, \| \mu \|, \|C\| = O(1)$


Under these assumptions, they provide a precise assymptotic caracterization of the eigenvalues and eigenvectors of the Hessian
$
H = \frac{\partial^2 L}{\partial w \partial w}(w).
$
More precisely:

1. They show that the empirical distribution of the eigenvalues of $H$ converges weakly almost surely to a probability measure $\mu$ defined by an equation on its Stieltjes transform.
2. When $\mu$ has a bounded support, they caracterize precisely the assymptotic postitions of potential isolated eigenvalues. When such isolated eigenvalues exists, the authors also compute the limiting angle between the correspondeing eigenvectors and the vectors $w,w_*,\mu$.
3. The authors illustrate their results on popular models and losses, showing the existence of interesting phenomena. In particular, they show that isolated eigenvalues can not only appears due to 'signal' in the data (large $\|\mu\|$) but also appear because of the model itself.


The main results are obtained through an so-called 'deterministic equivalent', that allows to study precisely the resolvant of the Hessian $H$.

**Limitations And Societal Impact:**

Yes

**Main Review:**

I found this paper very interesting and very relevant to a machine learning audience.

Obtaining a theoretial understanding of (non-convex) loss functions is an outstanding challenge that has recieved a lot of attention over the recent years. This paper provides a complete and precise analysis of the spectrum of the Hessian of G-GLM allowing to understand better the optimization of such models.
Even if this model remains simple it is very relevant because very popular and more realistic than the ones studied in the literature.
It would be interesting to see if one could extend this analysis to multilayer models, but one would probably have difficulties to obtain exploitable analytic formulas.

Hence I think that the approach taken by the authors is completely relevant. The results they obtain are a clear improvement of the ones present in the literature. It is interesting to see that some assumptions made in earlier papers are perhaps too simplistic (always predict a Marcenko Pastur eigenvalue density) for more realistic models such as G-GLM.

The results obtained in this paper are strong.
The paper is very well written, with many examples and comments that allow to understand the subtelties of the results.

The results of this paper can also be of practical interest, since one could use the top eigenvector of $H$ as initialization for iterative algorithm. On that point, I would be interested to know if one could achieve a correlation with $w_*$ by minimizing $H$ for some carefully chosen $\ell$ that is better than the spectral estimators proposed in [46].

I would be also curious to know if one could potentially obtain results about the Hessian at point $w$ that depend on the data $x_i, y_i$. One could for instance study the spectrum of $H$ for $w$ for which $L$ is small, but that may be very difficult...


To sum up, I think that this is a paper with strong results that answer and raises many interesting questions.


**Time Spent Reviewing:**

2h30

---

> ### Author Response · Authors · 2021-08-06
> **Response to Reviewer rM2S**
>
> We would like to thank the reviewer for his/her positive support and for the thorough and helpful remarks.
>
> * Question: On that point, I would be interested to know if one could achieve a correlation with $w_*$ by minimizing $H$ for some carefully chosen $\ell$ that is better than the spectral estimators proposed in [46].
> * Answer: The result proposed in [46] is "optimal" as long as the spectral method based on $H$ is used, for phase retrieval model and standard Gaussian data/sensing matrix with identity covariance. In the recent paper "Optimal combination of linear and spectral estimators for generalized linear models", the authors showed that improvement is possible by optimally combining linear and spectral (based on $H$-type matrix) estimators. Also note from our Theorem 3 that for generic Gaussian data $x_i \sim \mathcal{N}(\mu, C)$, the top eigenvector of $H$ is correlated to $C w_*$ instead of $w_*$. When interested in $w_*$ in the setting of $C \neq I$, one needs, to some extent, to estimate the covariance $C$ or the inverse $C^{-1}$ to recover the desired $w_*$, which is known to be non-trivial in the high dimensional (comparably large $n,p$) setting under study. Several possible workarounds are under investigation, but they seem to be out of the scope of this submission.
>
> * Question: I would be also curious to know if one could potentially obtain results about the Hessian at point $w$ that depend on the data $(x_i, y_i)$".
> * Answer: we thank the reviewer for pointing out this important and interesting setting. Indeed, one of our technical contributions is a "Gaussian decoupling" trick to explicitly express the dependence of $D$ (in the Hessian in Equation (4) of the submission) on the data matrix $X$ for G-GLMs, where we take $w$ to be *fixed* or *independent* of $X,y$. We conjecture that a similar trick can be applied if the dependence between $w$ and $X,y$ takes an explicit and simple form (e.g., $w$ is the least square solution to  $X,y$).

---

> > ### Comment · Reviewer_rM2S · 2021-08-15
> > **Response to the authors**
> >
> > Thank you very much for these explanations and this nice work. I advocate for acceptance.

---

### Decision · Program_Chairs · 2021-09-27

**Decision:**

Accept (Oral)

**Comment:**

This paper develops new mathematical results in random matrix theory that characterize the limiting spectral distribution of Hessian matrices corresponding to so-called generalized generalized linear models (G-GLMs) in the high-dimensional regime. Using the technique of deterministic equivalents, the authors are able to relax a number of the simplifying distributional assumptions that prior work has used to pursue analyses of this type. The result is an asymptotically exact characterization of the limiting spectrum that exhibits the types of structures that often show up in practice, including isolated outlier eigenvalues, multimodal densities, etc. As such, this paper offers the NeurIPS community a new and powerful perspective for understanding the spectra of high-dimensional models, and potentially paving the way for new developments in second-order optimization and beyond.

The phrase in the title of the paper, "more realistic," actually provides a good characterization of the scope and generality of the results: the focus here does not actually touch on realistic models, NNs, etc., but the G-GLM setup and the general distributional assumptions move the needle significantly in the "realistic" direction. While this paper does observe some phenomena that do tend to appear in practical configurations (such as large outliers), it remains unclear whether the explanations offered here are in fact the same ones that underlie the phenomena more generally. An expanded version of this paper would surely benefit from some detailed (empirical) analysis of this question. Still, absent these additions, the techniques and insights derived in this paper will nevertheless be of interest to the community, and this paper will make a great addition to NeurIPS.